# The 20-year impact of tobacco price and tobacco control expenditure increases in Minnesota, 1998-2017

**Michael V. Maciosek**[1]*, **Amy B. LaFrance**[1], **Ann W. St. Claire**[2], **Paula A. Keller**[2], **Zack Xu**[1], **Barbara A. Schillo**[3]

1 HealthPartners Institute, Minneapolis, Minnesota, United States of America, 2 ClearWay Minnesota, Minneapolis, Minnesota, United States of America, 3 Truth Initiative, Washington, District of Colombia, United States of America

* michael.v.maciosek@HealthPartners.com

**Data Availability Statement:** All simulation model inputs are provided in the manuscript and supplemental files.

## Abstract

### Introduction

Tobacco control programs and policies reduce tobacco use and prevent health and economic harms. The majority of tobacco control programs and policies in the United States are implemented at local and state levels. Yet the literature on state-level initiatives reports a limited set of outcomes. To facilitate decision-making that is increasingly focused on costs, we provide estimates of a broader set of measures of the impact of tobacco control policy, including smoking prevalence, disease events, deaths, medical costs, productivity and tobacco tax revenues, using the experience of Minnesota as an example.

### Methods

Using the HealthPartners Institute's ModelHealth™: Tobacco MN microsimulation, we assessed the impact of the stream of tobacco control expenditures and cigarette price increases from 1998 to 2017. We simulated 1.3 million individuals representative of the Minnesota population.

### Results

The simulation estimated that increased expenditures on tobacco control above 1997 levels prevented 38,400 cancer, cardiovascular, diabetes and respiratory disease events and 4,100 deaths over 20 years. Increased prices prevented 14,600 additional events and 1,700 additional deaths. Both the net increase in tax revenues and the reduction in medical costs were greater than the additional investments in tobacco control.

### Conclusion

Combined, the policies address both short-term and long-term goals to reduce the harms of tobacco by helping adults who wish to quit smoking and deterring youth from starting to smoke. States can pay for initial investments in tobacco control through tax increases and recoup those investments through reduced expenditures on medical care.

**Funding:** This study was funded by ClearWay Minnesota^SM, an independent nonprofit organization funded with three percent of Minnesota's tobacco settlement. ClearWay Minnesota works to reduce the harm of tobacco through research, action and collaboration and, as part of its work, funds and administers QUITPLAN Services (Minnesota's free quitline). All authors (ClearWay Minnesota, HealthPartners, and Truth Initiative) contributed to the study design. HealthPartners collected and analyzed the data, and led the interpretation of findings. All authors contributed to the writing of this report and decided to submit the article for publication.

**Competing interests:** AL, MM and ZX have received salary support through their employer for research projects with aspects of tobacco control funded by the U.S. Centers for Disease Control and Prevention, ClearWay Minnesota^SM, and the Robert Wood Johnson Foundation. All authors are employed by organizations whose missions include reducing the population harms of tobacco. All authors (ClearWay Minnesota, HealthPartners, and Truth Initiative) contributed to the study design. HealthPartners collected and analyzed the data, and led the interpretation of findings. All authors contributed to the writing of this report and decided to submit the article for publication.

## Introduction

Increasing tobacco price and investments in comprehensive tobacco control are high-impact evidence-based strategies for reducing smoking prevalence.[1, 2] To inform policy decisions, simulation models of United States (U.S.) populations have predicted policy effects on smoking prevalence, reductions in mortality, and medical costs.[2–9] For example, the U.S. Congressional Budget Office (CBO) simulated the 10-year impact of a federal cigarette tax increase, predicting a variety of outcomes that included mortality, medical costs, earnings, federal government revenue and federal spending.[9]

Most tobacco policy changes in the United States occur at state and local levels. State lawmakers may be motivated to support increases in tobacco taxes and/or tobacco control expenditures for various reasons, including improved health, reduced medial costs, improved productivity, and increased general revenue funds. Analyses of state-level policies have evaluated their impact on a variety of smoking-related behaviors,[1] and simulations have estimated the current and future impacts on prevalence, mortality, and medical spending,[10–14] but have not addressed the broader array of outcomes.

To expand the evidence base for formulating state-level tobacco control policy, we simulated the 20-year impact of tobacco price increases and increased investments in tobacco control in Minnesota from 1998 through 2017. From 1997 to 2016, adult smoking prevalence in Minnesota fell from 21.8% to 15.2%,[15] a 30% relative decrease. Part of this trend arose from earlier tobacco control policies, including the nation's first statewide clean indoor air law in 1975, the nation's first state-funded tobacco control program in 1985, and four state tax increases from 1985 to 1992. Significant policy changes after 1997 added to the trend, including strengthening the 1975 clean indoor air law to include all indoor public places, strengthening youth access laws, further increases in cigarette taxes and substantial increases in tobacco control expenditures. These changes were driven in part by ClearWay Minnesota^SM which was established through Minnesota's tobacco settlement to implement tobacco control programs and research using 3% of the state's tobacco settlement funds.[16] Since ClearWay Minnesota's inception, state tobacco control expenditures increased by as much as 10-fold, varying by year, while cigarette prices increased nearly 4-fold in response to settlement costs and increases in federal and state tobacco taxes. Large tobacco price increases are an effective known strategy to reduce tobacco use. Tobacco control expenditures support the state's free quit line, anti-tobacco media campaigns, tobacco education, communication, and community collaborations. Increased expenditures also support initiatives to reduce the harms of commercial tobacco to American Indian communities, and education and communication activities likely contributed to passing local clean air and Tobacco 21 laws and local restrictions on the sale of menthol cigarettes.

To provide a more comprehensive assessment of policy impacts, we used a detailed microsimulation model to simulate a broad range of outcomes, including smoking prevalence, disease events, mortality, smoking-attributable medical costs, productivity gains and state cigarette tax revenues within a 20-year timeframe. The results demonstrate the potential for health and economic gains in states that have not acted aggressively against tobacco.

## Methods

We simulated changes in policies using the HealthPartners Institute's ModelHealth^TM: Tobacco MN–a microsimulation model.[17] The model simulates annual changes in cigarette smoking behavior over the lifetimes of individuals and estimates the health and economic consequences of cigarette smoking. The simulation model, data inputs, and policy parameters are described in S1 and S2 Supplements, including details on the use of databases and literature to

inform the model. All planned model outcomes are reported in results. The model is capable of producing a wide range of intermediate and more granular outcomes such as number of quits and smoking initiations, lung cancer cases, and results by demographic factors that are not presented for brevity. The simulation does not include other tobacco products and the complex array of multiple product use that has developed in recent years.[18]

## Demographics and smoking status

We chose 1997 as our base year to assess 20 years of change, 1998–2017. We simulated 1.3 million individuals with age, sex, race/ethnicity, and educational attainment distributions representative of the Minnesota population of all ages in 1997.[19] Each simulated individual's insurance status and probabilities of changing insurance status over time varies with disability, employment and poverty status, as estimated for the U.S. from the U.S. Current Population Survey[19] and the Survey of Income and Program Participation.[20]

Youth smoking status is determined by its association with demographics as estimated from the first Minnesota Youth Tobacco Survey (MYTS) in 2000.[21] Year 2000 prevalence rates may produce a conservative baseline estimate of youth smoking in 1997, because national rates for youth were trending slightly downward at that time.[22] From MYTS we also estimated "net initiation" to represent the combined probability that one more youth becomes a smoker after accounting for quits during the year.

We estimated adult cigarette smoking status in 1997 from Minnesotans who responded to the 1996 or 1997 Behavioral Risk Factor Surveillance Surveys (BRFSS).[23] We combined two years of data to improve the precision of age- and sex-specific smoking behaviors, and in particular with cessation rates for which the sample is limited to current smokers. For ages 65 and older, we calibrated the initial smoking probabilities derived from BRFSS to be consistent with prevalence estimates from the first Minnesota Adult Tobacco Survey (MATS) in 1999.[24] We estimated cessation rates from combined 1996 and 1997 BRFSS data. Too few Minnesotans ages 18–24 were represented in BRFSS to yield reliable estimates. Therefore, we assumed that cigarette smoking status from ages 18 to 24 was the same as the model's predicted rates for 25-year-olds. This assumption could understate peak adult prevalence during the lifetime but provides a reliable estimate of prevalence prior to the ages of high harms from smoking-attributable chronic disease. To assign relapse probabilities, we constructed a curve based on literature that reveals a declining probability of relapse with greater time since quit.[25–29]

The number of cigarettes smoked per day was estimated by age, sex, educational status and race/ethnicity for the U.S. population and calibrated so the model reproduces cigarette packs smoked per capita in Minnesota in 1997.[30]

## Consequences of cigarette smoking

The model includes smoking-attributable cancers, and cardiometabolic and respiratory diseases identified in Smoking-Attributable Mortality, Morbidity, and Economic Costs (SAMMEC) estimates (S1 pages 33–46).[31] We obtained Minnesota death rates by age and sex for smoking-attributable conditions in 1996–1998 from Detailed Mortality Data.[32] We disaggregated these rates into never, current and former smokers using Minnesota adult smoking prevalence and relative risks of disease from the 2014 Surgeon General's Report.[31] We approximated Minnesota smoking-attributable non-fatal disease rates using the inverse of case-fatality rates for the United States in an update of the previously described U.S. version of ModelHealth™: Tobacco.[33]

Smoking-attributable medical costs measure the additional total cost of medical care of current and former smokers in excess of those of never smokers. To approximate smoking-

attributable medical costs of Minnesotans (S1 page 17), we scaled estimates for current smokers derived for the United States from 2000–2010 linked Medical Expenditure Panel Survey and National Health Interview Survey (MEPS-NHIS) data[34] using the ratio of Minnesota-to-U.S. per-capita healthcare expenditures.[35] We inflation-adjusted these costs to 2017 U.S. dollars using the Medical Care Consumer Price Index.[36] MEPS and other claims data include former smokers who quit smoking only after diagnoses that lead to increased healthcare utilization after quitting.[37–39] Therefore, for former smokers, we fit an exponential function to the relationship of current and former expenditures based on time since quit, using the relationship between current and former smoker mortality risks reported by the CBO (S1 page 16).[9]

In the model, productivity losses reflect absence from work,[40] lower productivity at work, [40] and lost years of work, including unpaid household productivity (S1 page 19).[41] We scaled U.S. measures of productivity by the ratio of Minnesota to U.S. per-capita earnings.[42]

## Simulation scenarios

We separately assessed the gains made by increasing investments in tobacco control expenditures in Minnesota and by increasing cigarette prices. We compared policy scenarios to the baseline scenario in which initiation and cessation rates are held constant at 1997 levels.

The baseline scenario predicts what the burden of cigarette smoking in Minnesota would have been if no policies, programs or trends had changed initiation and cessation rates from 1998 through 2017. This includes holding constant both changes aimed at reducing tobacco use and those aimed at increasing or maintaining tobacco use, such as tobacco industry marketing. Smoking prevalence falls over time in the baseline scenario, even with initiation and cessation rates held constant.

We had planned to analyze the effect of state clean indoor air legislation, but it was excluded because we were unable to simulate secondhand smoke exposure and its harms within the study timeframe. The analysis plan was not pre-registered.

**Increased Tobacco Control Investments (ITCI) scenario.**  In the 1997 baseline year, $0.81 per capita was appropriated for tobacco control in Minnesota from state, federal and foundation sources (S2 Supplement of Table S2.4),[43] expressed in 2017 dollars.[44] That amount grew to $9.85 per capita in 2001 before falling back to $4.07 by 2014[43] after state-controlled tobacco control endowments were used to balance the state budget in 2004. Nevertheless, an additional $376 million ($US 2017) was invested in tobacco control from 1998 through 2014 than would have if per-capita appropriations had remained at 1997 levels.

Comprehensive tobacco control programs coordinate multiple strategies, such as large- and small-scale media, school-based education, quit lines, and programs to decrease tobacco's accessibility to minors.[45] The effect of increasing investments in tobacco control in the simulation model is based on statistical studies that compare the association between tobacco control expenditures and smoking behaviors across US states. As described in detail in S2 Supplement, we reviewed studies that provide estimates of expenditure elasticities (the percent change in prevalence for each percent change in expenditure) that we could apply in the simulation model. The model estimates how annual percent changes to investments in tobacco control above 1997 levels [43] would modify the baseline initiation and cessation rates that define the baseline scenario. For the years 2015 to 2017, when appropriations data were not available at the time of analysis, we assumed that appropriations were maintained at 2014 per-capita levels.[43] We adjusted per-capita expenditures to 2017 dollars to remove inflation from the annual percent changes.

**ITCI + Price scenario.**  Minnesota average retail cigarette prices increased $0.76 per pack in 1999, the year after the state tobacco lawsuit settlement. In subsequent years, federal taxes

increased $0.77, and state taxes increased $3.10 unadjusted for inflation.[30] We created an ITCI + Price Scenario by adding to the ITCI Scenario changes in per-pack cigarette prices associated with tax increases and the conceptually similar increase in per-pack price following the 1998 Minnesota tobacco settlement.[16, 30] The difference in health and economic outcomes between this ITCI + Price Scenario and the ITCI scenario provides estimates of the impact of tax and 1998 settlement-associated price increases alone. Therefore, we assume that one policy does not impact the relative effectiveness on smoking behaviors of the other policy, either positively through synergistic effects or negatively through reducing the portion of remaining smokers or would-be smokers whose decisions are most amenable by policy.

Taxes in the model operate through price-prevalence elasticities (the percent change in smoking prevalence per one percent change in cigarette price), and price-intensity elasticities (the percent change in quantity smoked among continuing smokers per one percent change in price). These elasticities were drawn from literature with careful consideration of which studies provide the most appropriate price elasticity estimates as inputs to the simulation model, and on how the results of multiple studies should be combined to estimate price elasticity (see S2 Supplement). We assumed that tax increases have a one-time impact on adult cessation probabilities in the first year and a permanent impact on youth initiation. We inflation-adjusted each year's average tobacco price and each increase in price to 2017 U.S. dollars.

### Sensitivity analysis

We explored the influence of model parameters on adult prevalence, deaths, and medical care costs in sensitivity analysis. We explored the potential impact of any systematic bias in measuring cessation probabilities by 25% in either direction. Similarly, we examined the effects of systematic bias of up to 25% in the relative risks of disease of former and current smokers compared with never-smokers. Such bias may arise from the methods or data sources used to estimate the relative risk of disease for the U.S. population, or in applying those relative risks to the Minnesota population.

We explored systematic bias of up to 35% in estimating medical costs. Costs are based on MEPS, which produces lower estimates of health care costs per capita than National Health Expenditure Accounts (NHEA).[46] Therefore, the predicted cost-savings are more likely to be understated than overstated. We also explored the impact of alternative estimates of smoking costs derived to support the 2014 Surgeon General's Report[47] because they align more closely with NHEA costs.

We increased and decreased by 50% the estimates of expenditure and price elasticities. To compare tobacco control investments to resulting savings, we discounted the time sequence of investments in tobacco control and medical care costs at 3% per year.

Finally, we constructed the Price Increase Scenario for sensitivity analysis, in which the price increases associated with increased taxes and the 1998 tobacco settlement are added to the baseline scenario. We used this scenario to explore the impact of estimating increased investments in tobacco control marginal to price increases and to explore the impact of *not* estimating the impact of price increases marginal to increased tobacco control investments.

### Results

The model estimates that, due to increased tobacco control expenditures, 7,400 fewer youth and 139,000 fewer adults smoked cigarettes in 2017. Although investments fell after 2001, the impact on youth smoking stays at about 1 percentage point through 2017 (Fig 1; shown by the distance between the Baseline and ITCI scenarios for youth). In contrast the simulation estimates that adult prevalence continued to decline as a result of ITCI, even as investments

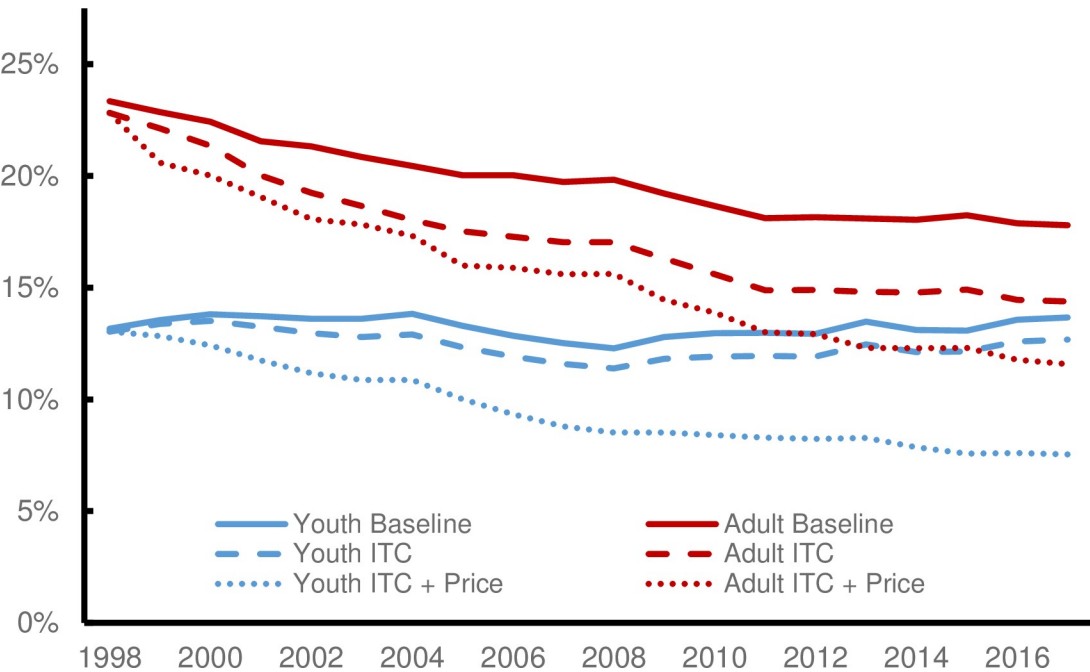

**Fig 1. Predicted trends in youth and adult prevalence in Minnesota under baseline and policy scenarios.**

declined from their peak, resulting in a 3.4 percentage point decline by 2017. This reflects additional quits among adults each successive year and additional non-smoking youth aging into the adult population.

The impact of the simulated price changes is shown by the difference between the ITCI and the ITCI + Price Increase scenarios (Fig 1). For youth, the estimated impact of price increases was greater than for investments in tobacco control. For adults, the opposite result was found.

Table 1 shows the 20-year cumulative outcomes for the main scenarios and the policy impacts. Annual results for each scenario are shown in S3 Supplement. The model estimates that $1.6 billion in smoking-attributable medical care costs was saved through 2017 as a result of increased investments in tobacco control. In addition, there were $1.2 billion gains in paid and unpaid productivity. The model estimates that these economic gains were accompanied by reductions in smoking-attributable cancer cases (4,100 fewer), combined cardiovascular and diabetes hospitalizations (24,600 fewer), respiratory disease hospitalizations (9,800 fewer) and deaths (4,100 fewer). If tax rates had stayed at 1997 levels, state tobacco tax revenues would have been an estimated $800 million lower over the 20 years in 2017 dollars as a result of reduced smoking prevalence that followed increased investments in tobacco control. However, the reduction in medical cost savings would have been twice as large as the lost tax revenues (see also S3 Supplement of Table S3.7).

The increases in tobacco prices reduced smoking-attributable medical care costs by an estimated $700 million and increased productivity by $900 million, while preventing 1,600 cancer cases, 9,300 cardiovascular and diabetes hospitalizations, 3,700 respiratory disease hospitalizations and 1,700 smoking-attributable deaths. Although tobacco price increases reduced cigarette sales, state tobacco tax revenues increased an estimated $3.2 billion in 2017 dollars over those 20 years, due to the increase in state tax rates behind some of the price increases.

The price changes following the tobacco settlement and tax increases had a larger impact on youth prevalence from 1998 to 2017 than did the increases in tobacco control expenditures

**Table 1. Cumulative impact of tobacco control policies, Minnesota 1998–2017.**

| Outcome | Scenarios | | | Policy impact | | |
|---|---|---|---|---|---|---|
| | Baseline | ITCI | ITCI + Price | ITCI[1] | Price[2] | Combined[3] |
| Youth smoking prevalence, ages 9–17[4] | 13.7% | 12.7% | 7.5% | -1.0% | -5.1% | -6.1% |
| Adult smoking prevalence, ages 18+[4] | 17.8% | 14.4% | 11.6% | -3.4% | -2.8% | -6.2% |
| Person-years of cigarette smoking, all ages | 16,803,500 | 14,754,100 | 13,084,000 | -2,049,400 | -1,670,100 | -3,719,500 |
| SA cancer cases | 174,700 | 170,600 | 169,000 | -4,100 | -1,600 | -5,700 |
| SA CVD and diabetes hospitalizations | 1,489,300 | 1,464,700 | 1,455,500 | -24,600 | -9,300 | -33,800 |
| SA respiratory disease hospitalizations | 442,900 | 433,200 | 429,400 | -9,800 | -3,700 | -13,500 |
| SA deaths | 186,000 | 182,000 | 180,300 | -4,100 | -1,700 | -5,700 |
| SA medical costs (millions of 2017 $US) | 27,900 | 26,300 | 25,600 | -1,600 | -700 | -2,300 |
| Productivity (millions of 2017 $US) | 4,808,300 | 4,809,500 | 4,810,400 | 1,200 | 900 | 2,100 |
| Cigarette tax revenues[5] (millions of 2017 $US) | 5,400 | 4,600 | 7,800 | -800 | 3,200 | 2,400 |

[1]ITCI Scenario compared to Baseline.

[2]ITCI+Price scenario compared to ITC.

[3]ITCI + Price Scenario compared to Baseline.

[4]Prevalence rates shown are for 2017; all other figures show cumulative estimates from 1998 to 2017.

[5]Tax revenues are shown using 1997 rates in the Baseline and ITCI scenarios (to show the effect of ITCI alone) and with actual tax rates in the ITCI + Price scenario. Cumulative revenues in the ITCI scenario using actual tax rates would be $10,400 million. SA = Smoking-attributable. CVD = Cardiovascular disease. ITCI = Increased Tobacco Control Investments.

(Fig 2), while the impact on adult prevalence was approximately the same in the two scenarios, indicating the potential for price changes to have greater impact in the future.

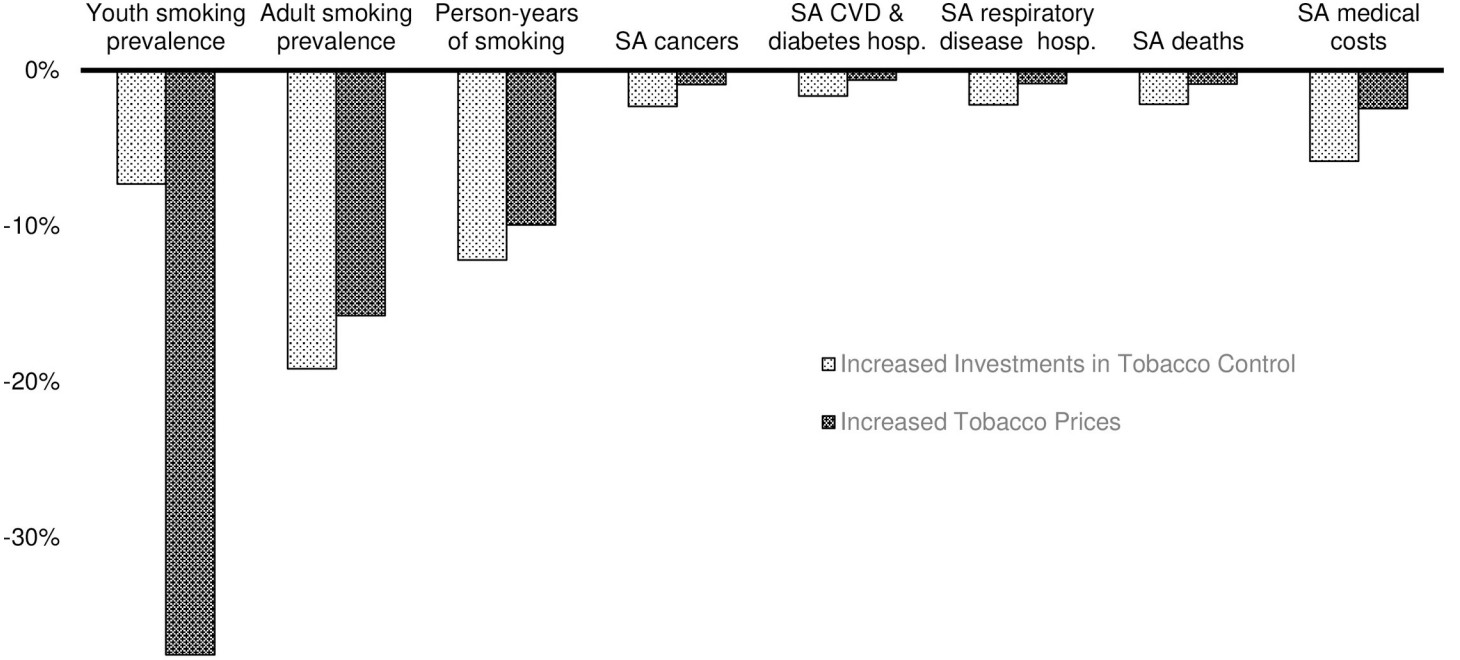

**Fig 2. Relative decline in tobacco use and harms by policy compared to baseline scenarios from 1998 to 2017 in Minnesota.** SA = smoking attributable. CVD—cardiovascular disease. hosp = hospitalization.

**Table 2. Sensitivity analysis.**

| Scenario | Increased tobacco control investments; ITCI compared to Baseline Scenario | | | Increased cigarette prices; ITCI + Price compared to ITCI Scenario | | |
|---|---|---|---|---|---|---|
| | Percentage point change in adult prevalence | Change in SA deaths | Change in SA costs (millions of 2017 $US) | Percentage point change in adult prevalence | Change in SA deaths | Change in SA costs (millions of 2017 $US) |
| **Base case** | **-3.4%** | **-4,100** | **-1,600** | **-2.8%** | **-1,700** | **-700** |
| Baseline cessation rates +25% | -3.1% | -3,500 | -1,500 | -2.8% | -1,800 | -800 |
| Baseline cessation rates -25% | -3.8% | -4,500 | -1,700 | -2.8% | -1,200 | -600 |
| Relative risk of SA disease +25% | -3.4% | -5,000 | -1,600 | -2.8% | -1,900 | -700 |
| Relative risk of SA disease -25% | -3.4% | -3,200 | -1,600 | -2.8% | -1,200 | -700 |
| SA medical costs +35% | -3.4% | -4,100 | -2,200 | -2.8% | -1,700 | -900 |
| SA medical costs -35% | -3.4% | -4,100 | -1,100 | -2.8% | -1,700 | -400 |
| Alternative SA medical costs | -3.4% | -4,100 | -2,200 | -2.8% | -1,600 | -900 |
| Effect of increased ITCI +50% | -4.4% | -5,100 | -2,100 | na | na | na |
| Effect of increased ITCI -50% | -2.3% | -2,800 | -1,200 | na | na | na |
| ITCI effect marginal to price effect | -2.9% | -4,000 | -1,600 | na | na | na |
| SA medical costs discounted 3% | -3.4% | -4,100 | -1,100 | na | na | na |
| Effect of increased prices +50% | na | na | na | -3.8% | -2,300 | -900 |
| Effect of increased prices -50% | na | na | na | -1.8% | -1,100 | -500 |
| Price effect not marginal to ITCI effect | na | na | na | -3.3% | -1,700 | -700 |

SA = Smoking-attributable. ITCI = Increased Tobacco Control Investments. na = not applicable

### Results of sensitivity analysis and secondary economic analyses

Changing baseline cessation rates by 25% changed the impact on adult prevalence in 2017 by 10% in relative terms compared to the base case (0.3 and 0.4 percentage points for decreased and increased baseline cessation rates, respectively), and had a similar relative impact on deaths and medical care costs prevented (Table 2). Changing the effectiveness of ITCI by 50% had a nearly proportional effect on adult prevalence in 2017, but less impact on deaths prevented and medical care costs saved. Using alternative cost estimates that align with those in 2014 Surgeon General's report increased savings by about 35%. Discounting the stream of medical costs to present value in 1997 reduced the measured costs prevented through 2017 by 34%, to $1.1 billion. The discounted additional investments in tobacco control over 1997 per capita levels were $270 million over 1998 to 2017 (S3 Supplement of Table S3.7). Therefore, discounted savings realized through 2017 were 3 times greater than additional investments made through 2017. In addition, discounted medical costs saved remain greater than discounted investments even when the effectiveness of additional investments is reduced by 50%.

## Discussion

The simulation results suggest that from 1998 to 2017, the level of investments in tobacco control reduced tobacco harms in Minnesota more than did the increases in the price of cigarettes that occurred in the state. Minnesota's additional investments in tobacco control reduced smoking-attributable deaths by an estimated 4,100 and smoking-attributable medical costs by $1.6 billion. Medical cost savings were twice the size of the reduction in tax revenue that were attributable to declining cigarette sales with increased tobacco control investments. Over the same period, Minnesota's price increases reduced deaths by an additional 1,700 and reduced medical costs by $700 million. With increasing tax rates, cigarette tax revenues increased more than the expenditures on tobacco control despite declining sales attributable to both increased tobacco control expenditures and cigarette prices.

The health impacts from increased investments in tobacco control seem disproportionate to those of price increases given their relative impact on adult prevalence in 2017. However, the simulation estimates that the increased investments in tobacco control had a larger impact on adult prevalence in early years (Fig 1). Therefore, within the 20-year period, increased investments in tobacco control accumulated more years of people living as former smokers and a longer average time since quit. Minnesotans who were younger than 18 anytime between 1998 and 2017 remain decades away from their peak risk of smoking-attributable disease, so the large impact of price increases on youth prevalence did not translate into larger reductions in the harms of tobacco by 2017.

Our time horizon does not capture all of the benefits of changes between 1998 and 2017. If the analyses were projected into the future, we would see greater benefits as youth who avoided smoking initiation become adults, and as adult former smokers place more years between their last cigarette and their current health risk. After 2017, the impact of price changes could exceed those of increased investments in tobacco control.

The range of outcomes reported here highlight the impact of tobacco control policy and its relevance to the varied interests of individual legislators and their constituents. Most prior studies of state tobacco policy report effects on prevalence and mortality, and less frequently, medical costs. This study adds effects on disease events, productivity, and state cigarette tax revenues to the literature. For Minnesota, Levy et al. estimated that price increases during 1994 to 2011 reduced smoking prevalence by 13.5% and 13.6%, in relative terms, for men and women ages 15 and above, respectively.[10] In our analysis, smoking prevalence among adults decreased 15.7% in relative terms in response to price changes from 1997 to 2017 (Fig 2). The difference in estimates may be driven largely by the $1.60 tax increase in 2013 and subsequent inflation indexing of state tobacco taxes at the end of the time period. Levy et al. also estimated the impact of tobacco control expenditures inclusive of media campaigns. The relative declines in prevalence they estimate for 1994 to 2011, 6.2% for men and 5.6% for women, were lower than our estimate for adults from 1997 to 2017, 19.1% (Fig 2). Levy et al. estimated expenditures in categories of intensity rather than as a continuous variable and used effectiveness estimates corresponding to those categories rather than expenditure elasticities. Differences in model structure also are likely to contribute to differences in results.

We previously estimated the health and economic gains that Minnesotans realized between 1998 and 2017 from reductions in tobacco prevalence during that period, without regard to the source of prevalence decline.[17] The ITCI + Price Scenario in the current study produced a lower adult prevalence in 2017 (11.6%) than was used to estimate the impact of all reductions in prevalence regardless of cause in the prior report (13.5%). This seemingly incongruent result may have several sources. Tobacco industry marketing efforts to counter the impact of investments in tobacco control and tax increases may have prevented initiation rates from declining

and cessation rates from rising as much as they otherwise would have.[48, 49] The tobacco industry spent an average of $165 million on tobacco marketing in Minnesota each year from 1998 to 2017 as estimated from national expenditures and state tobacco sales.[50] The price and/or expenditure elasticities we derived from literature could be biased upward. While there is extensive literature indicating that both investments in tobacco control and increasing prices reduce tobacco use, the size of the reductions obtained from literature may be imprecise. In particular, model results by age group (not shown), indicate that prevalence is lower in the ITCI scenario than actual prevalence rates between ages 25 and 64, but not in younger or older adults. The effect sizes from the literature may not generalize to the size of additional tobacco control investments or tobacco price increases experienced in Minnesota between 1998 and 2017, or the Minnesota population may have responded differently. In addition, our effect sizes are mathematically adjusted to avoid double counting of relapse as described in S2 Supplement and the proper degree of adjustment is uncertain. Finally, the model assumes relapse rates are equal for all quits due to a lack of data and technical limitations to assigning a reason for each quit. Policy-induced quits could have a higher relapse rate, in which case the simulation would somewhat overstate long-term policy effects.

Other limitations should also be noted. The simulation does not include all harms of smoking, including direct costs outside the medical care sector, and therefore may understate the benefits of reduced tobacco use. For example, despite a requirement that only 'fire safe' cigarettes be sold in the state, in 2018 careless smoking was faulted in 10 of 30 fire fatalities with known cause in Minnesota.[51] In the U.S., smoking-related fires were responsible for $1.36 billion in direct costs such as property damage and firefighting expenses, and $1.16 billion in productivity losses in 1995 (the most recent comprehensive estimate we found).[52] Higher cigarette prices are associated with fewer residential fires and fire deaths.[53] Our estimates also exclude benefits of reducing secondhand smoke which contributes to cancer, respiratory disease, and cardiovascular disease, low birth weight, and their associated medical costs.[31] Max et al. estimated that the portion of attention deficit hyperactivity disorders associated with secondhand smoke costs the U.S. health care system $2 billion and the U.S. education system $9 billion when they used serum cotinine levels as a biomarker for secondhand smoke exposure.[54]

The simulation also excludes the impact of policies on use of tobacco products other than combustible cigarettes. Expenditures on tobacco control are likely to reduce other forms of tobacco use, which would cause us to understate the total benefits of ITCI. While Minnesota taxes all forms of tobacco, including e-cigarettes, higher taxes on cigarettes (other taxes held constant) might lead tobacco users to substitute other forms of tobacco. While other tobacco products may produce less harm, they are not without risk, and excluding them may cause us to slightly overstate the impact of cigarette price increases.

S1 and S2 Supplements detail model inputs, including the literature reviewed, decisions on study inclusion and how study results are used in the model and database analyses. Like all simulations, ours is limited by the precision of model inputs, such as using self-reported tobacco use and making adjustments to US data on costs and productivity to approximate Minnesota-specific values. Our assumption on tobacco use for adults less than 25 years of age would cause us to simulate too few former smokers if peak adult prevalence occurs before age 25. The most uncertain inputs to the model are the elasticity estimates for increased investments in tobacco control and increased cigarette prices. We used a broad range of values in sensitivity analysis to reflect the uncertainties noted above. Although estimating return on investment for ITCI was not a study goal, sensitivity analysis results indicate that even at lower estimates of effectiveness, investments in tobacco control were more than offset by reductions in smoking-attributable medical expenditures, indicating a positive return on investment

during the limited time horizon, even when the value of improved health and productivity gains were excluded from the calculation. The additional state tobacco tax revenues collected during this period were greater than additional investments in tobacco control, even though the price increases reduced cigarette sales.

## Conclusion

This study enhances our understanding of the impact of two key policy levers identified by the CDC as best practices for U.S. states to reduce the harms of tobacco–increasing tobacco prices and increasing investments in tobacco control.[55] Investments in tobacco control can more than pay for themselves in the mid to long-term. Although Minnesota did not fund investments in tobacco control through a tax increase, the state's experience indicates that revenues from a tax increase can fund the upfront investments even as these policies reduce cigarette packs sold. Strategically, the combination of the policies targets both short-term and long term harms of tobacco by supporting established smokers in their quit attempts while deterring youth from starting smoking through price increases and youth-focused strategies such as media campaigns and reducing youth access. Therefore, policy makers who seek to improve health and reduce costs would be wise to consider aggressively implementing both of these policies. Comprehensive state tobacco control programs that combine price increases, investments in tobacco control and policies that prevention initiation, and help smokers who want to quit have been found to be the most effective strategy for reducing the harms of tobacco.[1, 55]

## Supporting information

**S1 Supplement. HealthPartners Institute ModelHealth[TM]: Tobacco MN model documentation model version 1.2.**
(DOCX)

**S2 Supplement. Background and model implementation of policies for Minnesota tobacco policy analyses.**
(DOCX)

**S3 Supplement. Annual results tables.**
(DOCX)

## Author Contributions

**Conceptualization:** Michael V. Maciosek, Ann W. St. Claire, Barbara A. Schillo.

**Formal analysis:** Michael V. Maciosek, Zack Xu.

**Funding acquisition:** Michael V. Maciosek, Amy B. LaFrance, Barbara A. Schillo.

**Investigation:** Michael V. Maciosek, Amy B. LaFrance, Ann W. St. Claire, Zack Xu.

**Methodology:** Michael V. Maciosek, Zack Xu, Barbara A. Schillo.

**Project administration:** Amy B. LaFrance, Ann W. St. Claire, Paula A. Keller.

**Software:** Zack Xu.

**Supervision:** Michael V. Maciosek, Ann W. St. Claire, Paula A. Keller.

**Validation:** Michael V. Maciosek, Zack Xu.

**Writing – original draft:** Michael V. Maciosek, Amy B. LaFrance, Ann W. St. Claire, Paula A. Keller.

**Writing – review & editing:** Michael V. Maciosek, Amy B. LaFrance, Ann W. St. Claire, Paula A. Keller, Zack Xu, Barbara A. Schillo.

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
