## [Decision Letter · Decision Letter 0]

3 Jan 2020

PONE-D-19-28016

The 20-year impact of tobacco price and tobacco control expenditure increases in Minnesota, 1998-2017

PLOS ONE

Dear Dr. Maciosek,

Thank you for submitting your manuscript to PLOS ONE. After careful consideration, we feel that it has merit but does not fully meet PLOS ONE’s publication criteria as it currently stands. Therefore, we invite you to submit a revised version of the manuscript that addresses the points raised during the review process.

We would appreciate receiving your revised manuscript by Feb 17 2020 11:59PM. To enhance the reproducibility of your results, we recommend that if applicable you deposit your laboratory protocols in protocols.io, where a protocol can be assigned its own identifier (DOI) such that it can be cited independently in the future. For instructions see: http://journals.plos.org/plosone/s/submission-guidelines#loc-laboratory-protocols

We look forward to receiving your revised manuscript.

Kind regards,

Stanton A. Glantz

Academic Editor

PLOS ONE

Journal Requirements:

[AL, MM and ZX received funding through a research contract between their employer and Clearway Minnesota (http://clearwaymn.org). Coauthors from Clearway Minnesota participated in study design and manuscript preparation.]  

We note that one or more of the authors are employed by a commercial company: 'Clearway Minnesota'.

3. Please include a new copy of Table 1 in your manuscript; the current table is difficult to read. Please follow the link for more information: http://blogs.PLOS.org/everyone/2011/05/10/how-to-check-your-manuscript-image-quality-in-editorial-manager/

Additional Editor Comments (if provided):

As you revise the paper, please pay particular attention to the suggestions that would make the results and implications clearer to policy makers. At the same time, including more of the analytical details in the body of the paper can also be accommodated, so long as there are clear statements of the policy implications in the abstract, introduction and discussion highlighted with appropriate subheads.

Like most work in this area, the paper concentrates on health cost savings. Reductions in tobacco use also have important cost implications for other sectors of government and society in general. For example, Wendy Max's group has shown that the cost savings to the educational system by reducing AHDH related to secondhand smoke exposure among youth are substantially larger than the direct medical costs (Childhood secondhand smoke exposure and ADHD-attributable costs to the health and education system. Max W, Sung HY, Shi Y. J Sch Health. 2014 Oct;84(10):683-6. doi: 10.1111/josh.12191). There are other issues, such as fires that are also impacted. I am not suggesting that the authors expand the scope of their paper, but they should at least mention these other economic impacts because they are important to policy makers and show that the estimates in their paper are something of a lower bound.

The authors need to recognize that their numbers are estimates and report their results rounded to 2 or 3 significant digits, based on the precision of the input variables in the model.

Reviewers' comments:

Reviewer's Responses to Questions

**Comments to the Author**

1. Is the manuscript technically sound, and do the data support the conclusions?

Reviewer #1: Yes

Reviewer #2: Yes

Reviewer #3: Yes

2. Has the statistical analysis been performed appropriately and rigorously? 

Reviewer #1: Yes

Reviewer #2: Yes

Reviewer #3: Yes

3. Have the authors made all data underlying the findings in their manuscript fully available?

Reviewer #1: Yes

Reviewer #2: Yes

Reviewer #3: Yes

4. Is the manuscript presented in an intelligible fashion and written in standard English?

Reviewer #1: Yes

Reviewer #2: Yes

Reviewer #3: Yes

5. Review Comments to the Author

Reviewer #1: This study assessed the 20-year impact of tobacco price increases and increased tobacco control expenditure during 1998-2017 in Minnesota using a microsimulation model developed by the authors. Different from most of similar studies in the literature, this study include a broader array of outcome: smoking prevalence, disease events, mortality, smoking-attributable medical costs, productivity gains, and state cigarette tax revenues. This a well-written article with details of their models and simulation methodologies presented in the appendices. Although this study made many assumptions about their model parameters, it provided sensitivity analyses to explore the influence of a wide range of changes in model parameters. This study demonstrated that increasing tobacco prices and investments in tobacco control simultaneously is effective to reduce the harms of tobacco use and is cost-saving in the mid- to long-term. I only have several minor comments.

1. Page 7, the methodology of estimating the consequences of cigarette smoking was not explained sufficiently. More description is needed regarding whether and how an econometric model was developed to estimate the smoking-attributable medical costs for current smokers. According to Table S3.1 under the baseline scenario, smoking-attributable medical costs was $1.324 billion in 2017. It would be great if the authors can add some discussion to compare their annual SA medical cost estimates with similar studies in the literature.

2. Page 9, lines 151-153: It is not clear whether the price increases mentioned here ($0.76, $0.67, and $3.10) reflect constant dollar or nominal term. For example, Line 152 indicates that in subsequent years after 1999, the federal excise tax on cigarettes increased by $0.67 per pack. However, during that period, the federal cigarette taxes increased 3 times: by $0.10 per pack in 2000, by $0.05 per pack in 2004, and by $0.61 per pack in 2009. The sum of these increases is $0.76 which does not match to $0.67.

3. Page 9, lines 155-157: “The difference in health … between this ITC + Price Scenario and the ITC scenario provides estimates of the impact of tax increases alone”. This sentence is not correct because this difference also reflects the impact of the price increase due to the Minnesota tobacco lawsuit settlement.

4. Page 11, lines 188-189. The reduced numbers of smokers in 2017 (7,434 for youth and 138,123 for adults) due to ITC do not add up to the number shown in Table S3.4 (146,402).

5. Page 11, lines 201-205; page 12, line 206 and lines 211-203: All the numbers presented in these sentences do no match to the numbers shown in Table 1 and Tables S3.4.

6. Page 12, line 207 and line 215: Add “(data not shown”) after “$784 million”, and similarly after “$3.20 billion”.

7. Page 13, the last row “Base case”: the number shown in the last column does not match to the corresponding cell in Table 1.

Reviewer #2: This manuscript presents the findings of a microsimulation to assess the impact of tobacco control policies, including tax increases and increased expenditures on tobacco control, in Minnesota during the twenty year period spanning 1998-2017. The study found that that increased expenditures on tobacco control prevented 37,421 cancer, cardiovascular, diabetes and respiratory disease events and 3,839 deaths over 20 years. Additionally, increased prices prevented an additional 14,702 additional events and 1,662 deaths. The authors conclude that both the net increase in tax revenues and the reduction in medical costs were greater than the additional investments in tobacco control, and that states can pay for initial investments in tobacco control through tax increases and recoup those investments through reduced healthcare expenditures.

This manuscript is generally well written, the analytic approach is free from any major methodological pitfalls, and would make a meaningful contribution to the scientific literature. Nonetheless, the manuscript could be improved, including more prominent recognition of various limitations of the analysis. Specific recommendations for revision are as follows:

1. Abstract. Page 2. Introduction. The first sentence of the abstract understates the considerable body of scientific evidence on the efficacy of tobacco control programs; specifically, the language needs to more clearly state the direction of the effect. They don’t just “influence” tobacco use and related health and economic outcomes – they reduce these outcomes.

2. Introduction. Page 4. Third paragraph. The third sentence references “prior initiatives” and “significant policy change during this period.” However, it’s not clear to a lay reader what this means, particularly those without tobacco control expertise. A sentence should be added that clearly articulates what policies were actually implemented.

3. Introduction. Page 4. Since the manuscript only focuses on tobacco control program expenditures and price increases (i.e. and not other specific proven strategies such as smoke-free policies), it would be helpful to better set the stage for inclusion of these variables in the Introduction, including the focus on increased price. One way to accomplish this would be to reinforce that tobacco control expenditures support a variety of efforts, including state and community programs that influence policy adoption, and that increasing price has previously been noted as the single most effective intervention for reducing consumption. That way you better set the stage for the reader around the measures focused upon in the manuscript.

4. Introduction. Page 5. First paragraph. The last two sentences of the Introduction section are implication and discussion points, not Introductory content. They should be deleted here, and reinforced in the Discussion section.

5. Methods. Page 5. Demographics and smoking status. It’s not clear what specific age groups from the “Minnesota population in 1997” the data were adjusted to – was it all individuals, those 18+, or something else?

6. Methods. Page 6. The paragraph beginning “we estimated adult cigarette smoking” lacks necessary precision for the reader to understand the rationale for the employed analytic approach. For example, it’s not clear why 1996 and 1997 data were combined (sample size issue?). Additionally, it’s not clear what is meant by “calibrating the initial smoking probabilities”. Furthermore, it’s not apparent how the 18-24 year old age group, for which there was insufficient sample, could be predicted using estimates for 25-year olds, which reflects only a single age year; did you mean some range that began with 25 as the lower bound? If sample size were an issue, it would seem that you could consider adding multiple years of data to afford sufficient sample for 18-24 year olds, which is a critical smoking initiation demographic. If you can’t this needs to be duly noted as a limitation of the analysis.

7. Methods. Page 7. Consequences of Cigarette Smoking. More clarity needs to be provided in the first paragraph of this section to articulate what smoking-attributable diseases were included. The current framing suggests it could be some, but not all. To clarify, it might be more prudent to place the onus of inclusion on outcomes found to be significantly associated with smoking in the 2014 Surgeon General’s Report, rather than focusing on SAMMEC, which is no longer in existence.

8. Methods. Page 8. It’s unfortunate that the authors could not include smoke-free policies in the analysis, which was undoubtedly a contributor to both declining secondhand smoke exposure, as well as reduced smoking, in Minnesota during the assessed period. This needs to be stated as limitation of the manuscript (along with other proven strategies that couldn’t be included). Additionally, the text in this paragraph of the Methods should use the more commonly recognized term of “secondhand smoke exposure” instead of “secondary smoke exposure”. It’s also not clear what is meant by the statement “the analysis plan was not pre-registered.”

9. Methods. Page 8. It’s strongly recommended that the authors identify another abbreviation than “ITC” for “Increased Investments in Tobacco Control.” The abbreviation “ITC” has long been associated with the International Tobacco Control Survey in the tobacco control field, and thus, using this abbreviation may cause confusion when findings are reported elsewhere or taken out of the direct context of this study. Instead, authors could consider something like Increased Tobacco Control Investments, or “ITCI”.

10. Methods. Page 9. It’s questionable whether it is safe to assume that one policy does not impact the relative effectiveness on smoking behaviors of the other policy. Increased expenditures for tobacco control programs include state and community interventions, as outlined in CDC’s Best Practices for Comprehensive Tobacco Control Programs. Price increases are considered a product of those programs. As such, increased expenditures for programs would be associated with increased momentum around pricing strategies like excise taxes. Since there’s not much else the authors can do post hoc, at the least this should be duly noted as a limitation of the analysis.

11. Discussion. Page 14. It is strongly recommended that the authors not pit individual interventions against one another. The framing around investments being more impactful than price would surely be used by opponents of cigarette excise taxes to state that they aren’t needed because they aren’t as effective. The major take home here is really that both are effective and important as part of a comprehensive approach to tobacco prevention and control at the state level.

12. Discussion. Page 15. The authors appropriately note that tobacco industry marketing efforts could counter the impact of tobacco control investments. However, this point could be made clearer for lay readers if the most recent data from the Federal Trade Commission were cited. In 2017, $9 billion was spent on advertising and promotion of cigarettes— more than $1 million every hour.

13. Discussion. Page 16. The manuscript needs a paragraph that clearly articulates the limitations of the employed analytic approach, including several of the items noted previously in this review (e.g. lack of smoking estimates for 18-24 year olds, omission of full scope of policies such as smoke-free, assumption about independent effect of price and tobacco control program expenditures, etc).

14. Discussion. Another key point to reinforce in the Discussion is that this analysis only accounts for cigarette smoking, not all tobacco products, including other combustible products, smokeless tobacco products, and e-cigarettes. During the later part of the assessed period, particularly from 2011-2017, massive increases in the use of e-cigarettes were observed among youth – both nationally and in Minnesota. Therefore, this model doesn’t account for the full scope of tobacco products being used, as well as associated health risks/costs. Cigarette smoking is certainly responsible for the overwhelming burden of death and disease caused by tobacco use, but these other products are not without risk. Accordingly, this needs to be explicitly stated as a limitation of the analysis, and ideally, a reference to the diversification of the tobacco product landscape should also be made in the narrative elsewhere in the manuscript.

15. Discussion. Another essential point that is briefly referenced, but should be more prominently focused upon, in the manuscript is that a comprehensive approach is warranted. There is no single panacea, and for several decades, tobacco control has most effectively functioned through a comprehensive lens that addresses the various drivers influencing initiation and cessation. As currently framed, the manuscript could be misinterpreted as suggesting that either expenditures and price are effective alone. It’s also important to reinforce that it’s not just making the expenditures, but what those dollars are actually used for; large amounts of dollars allocated to ineffective initiatives or strategies will do little for preventing initiation and reducing consumption at the population level. The expenditures needs to be focused of proven strategies for which evidence of efficacy and cost-effectiveness have been established.

Reviewer #3: This is an important paper with findings relevant to continuing policy decisions about the investment of millions of tobacco settlement dollars. The economic analyses are detailed and comprehensive, and provide the basis for important policy discussions and conclusions. As such, the strength and details of these analyses are needed. However, this leads to a key question: what is the primary audience of this paper? The decision appears to be health policy makers, and not health economists. The details in the Supplemental Files are dense but important, and if more were included in the paper it would become more of a health economist paper. However, as written, many of the critical technical details of the model, parameter definitions and selection, sensitivity analysis, and technical aspects of the results are left to the Supplemental Files. Even so, the language of the paper and topics discussed still focus too much on economic issues rather than presenting key findings most relevant to the policy makers. For example, while noted in the discussion that "estimating return on investment for increased ITC was not a study goal..." (lines 290-291), the findings of a positive return on investment even during the limited time horizon is very important, and should not be buried deep in the Discussion. Similarly, the sensitivity analysis is important to these potential return on investment discussions and conclusions, but as presented, these analyses are written more for economists, and as such, will be too confusing to most policy makers. The critical link between the sensitivity analyses and return on investment conclusions needs to be clearly defined when introducing them (line 226). Also, while the Figures (1 and 2) are good at communicating the impact of ITC and Price, the specific data in Table 1 and Supplemental Files S3 are not presented well in the text for policy makers, and especially not summarized well in Abstract. The focus is on deaths and SA incidents averted; however, the bottom line for many policy makers is dollars saved. The medical cost savings and productivity gain estimates are unique to these analyses and should be given primary emphasis, along with the resulting return on investment calculations based upon them.

Also, the wealth of literature review, technical discussions, and econometrics in Supplemental Files S1 and S2 are very impressive. As noted above, if the primary audience of the paper is health policy makers, then these details need to stay there. However, the descriptions of the technical approaches in the main paper should be reviewed and considered: are they communicating the technical quality of the work presented in the Supplemental Files to the policy makers as effectively as possible? In many places, the reader should be specifically referred to the excellent presentation of the issues and parameter selection criteria in S1 and S2.

More detailed comments:

Lines 43-48: I believe this is the only microsimulation model -- and the others, particularly the Levy model is a relatively crude macrosimulation model. This distinction should be emphasized.

Lines 59-65: the potential of these findings to provide data needed for cost-effectiveness and return on investment calculations should be introduced and emphasized.

Line 70: the details in literature review supporting selection of data inputs, policy parameters, and details in model should get more emphasis here.

Lines 128-130: The discussion of clean indoor air legislation should be revised. The specific impact on CIA on health effects of reduced exposures to secondhand smoke are not included (as noted); however, the policy impact of CIA on other outcomes by default are included in the ITC. While these are not specifically included in the model parameters, their impact is captured in the overall results. This should be noted.

Lines 131-137: The details of annual tobacco control investments are provided in Supplemental Files. Guide readers to these details. This also could apply to other data included in the model and only shown in detail in Supplemental Files; i.e., annual excise tax levels and average annual retail cigarette prices.

Lines 167+: as noted above, the important relevancy of these simulations for explaining return on investment after adjusting for possible biases and reduced parameter estimates for impact of ITC or price, i.e., expenditure and price elasticities, etc.

Line 192: "...due to lagged effect of prior investments." will not be understood my almost all readers as being primarily due to two model decisions: first, only one time impact of price increases applied at time of tax increase (standard econometrics) for adults, but impact on youth extended; and second, the use of cumulative funding with 25% discount (a la Farrelly et al models). Maybe this is too technical a point to discuss in results, but might merit being raised in discussion.

Line 209: the detailed impacts are shown in S3, but more of them could be shown in an extended Table 1. Additionally, details on annual ITC should be linked with results in S3 to highlight return on investments. Also, S3 could show more details on annual state tax revenues under Baseline Scenario as well as under actual tax increases. Finally, it would be outstanding to see a combined return on investment table combining tax revenues, ITC investments, and SA medical costs. Such a table could have very important policy implications.

Table 1: something does not look right. I did not search out all the details in Supplemental Files, but the deaths do not look right. SA cancers (incidents) and SA deaths are almost identical; however, SAMMEC estimates are that only about one-third of SA deaths are due to cancer. This needs to be checked, since SA deaths should be much higher than SA cancer incidents and deaths.

Figure 2 (lines 217-224): emphasis on percent changes, but policy impact in $s only shown in Table 1. Figure could be 2a (current) and 2b with numbers (number of youth prevented, increase number of adult quitters, SA incident cases prevented, SA deaths averted, and most importantly, SA costs saved (in $s), productivity gains (in $s) and (to be computed) total savings (SA medical averted and productivity gains).

Lines 225-237: as noted above, the impact of sensitivity analyses on return on investment calculations needs more emphasis. Lines 233-236 briefly discuss these findings; but, these calculations need much more emphasis and should be provided in greater detail; maybe even added as additional columns in Table 2.

Lines 239-255: this discussion is relevant, but may not strike the right balance. Make it clear from the first sentence that both ITC and price increases are needed: they are complimentary in timing and level of impacts produced.

Lines 256-267: discussion of Levy et al model results relevant, but this is a microsimulation model that differs in many ways from their model beyond parameter estimates.

Lines 287-296: discussion of uncertainties and imprecisions in model parameters and inputs is always relevant. The details in S1 and S2 provide very good discussion of how parameters and model specifications were done. Point readers to these discussions and reviews. Yes, uncertainties about elasticities, especially about ITC, limit model predictions. Thus, the need for good sensitivity analyses (as was done). Nevertheless, the positive return on investment calculations were found, as well as the positive revenue gains for tax increases (which need more emphasis with more specific numbers provided, even thru this has been documented in many prior econometric analysis).

FINALLY, it is suggested that future analyses estimate the even greater impacts (on all outcome measures, but particularly SA cost savings) if ITC levels would have been higher (like full BP levels or even 50% of BP level) or would not have been reduced in 2004.

6. PLOS authors have the option to publish the peer review history of their article (what does this mean?). If published, this will include your full peer review and any attached files.

Reviewer #1: No

Reviewer #2: No

Reviewer #3: No

---

## [Author Response · Author response to Decision Letter 0]

18 Feb 2020

We appreciate the reviewers and editor volunteering their time to help us improve the manuscript. We discuss each suggestion below.

Additional Editor Comments (if provided):

As you revise the paper, please pay particular attention to the suggestions that would make the results and implications clearer to policy makers. At the same time, including more of the analytical details in the body of the paper can also be accommodated, so long as there are clear statements of the policy implications in the abstract, introduction and discussion highlighted with appropriate subheads.

We appreciate that the editor recognized the conflict between suggestions to focus the paper and requests to add both technical details and further policy implications. As suggested, we have added both detail and policy statements rather than having a more focused paper. To keep a bound on the technical detail within the manuscript, we now refer readers to places in the supplements where they can find additional detail. 

Like most work in this area, the paper concentrates on health cost savings. Reductions in tobacco use also have important cost implications for other sectors of government and society in general. For example, Wendy Max's group has shown that the cost savings to the educational system by reducing AHDH related to secondhand smoke exposure among youth are substantially larger than the direct medical costs (Childhood secondhand smoke exposure and ADHD-attributable costs to the health and education system. Max W, Sung HY, Shi Y. J Sch Health. 2014 Oct;84(10):683-6. doi: 10.1111/josh.12191). There are other issues, such as fires that are also impacted. I am not suggesting that the authors expand the scope of their paper, but they should at least mention these other economic impacts because they are important to policy makers and show that the estimates in their paper are something of a lower bound.

We have added the following paragraph to the discussion section that speaks to the excluded economic and other benefits that are not included in the model: 

“Other limitations should also be noted. The simulation does not include all harms of smoking, including direct costs outside the medical care sector, and therefore may understate the benefits of reduced tobacco use. For example, despite a requirement that only ‘fire safe’ cigarettes be sold in the state, in 2018 careless smoking was faulted in 10 of 30 fire fatalities with known cause in Minnesota.(51) In the U.S., smoking-related fires were responsible for $1.36 billion in direct costs such as property damage and firefighting expenses, and $1.16 billion in productivity losses in 1995 (the most recent comprehensive estimate we found).(52) Higher cigarette prices are associated with fewer residential fires and fire deaths.(53) Our estimates also exclude benefits of reducing secondhand smoke which contributes to cancer, respiratory disease, and cardiovascular disease, low birth weight, and their associated medical costs.(31) Max et al. estimated that the portion of attention deficit hyperactivity disorders associated with secondhand smoke costs the U.S. health care system $2 billion and the U.S. education system $9 billion when they used serum cotinine levels as a biomarker for secondhand smoke exposure.(54)”

The authors need to recognize that their numbers are estimates and report their results rounded to 2 or 3 significant digits, based on the precision of the input variables in the model.

We have rounded results to 2 digits because differences between the results of scenarios in sensitivity analysis that test substantial changes in model parameters are not visible when rounded to 3 digits. This is an indication that model precision may be better than 3 digits, and rounding to 2 significant digits preserves the utility of the sensitivity analysis. Please note we reviewed all numbers in the text given that Reviewer 1 noted several inconsistencies between the numbers in the text and the tables. In all cases, the numbers in the tables were found to be correct. Therefore, changes to some numbers in track changes are a combination of rounding to 2 significant and corrections. These track changes may give the appearance of incorrect rounding as any changes to numbers in the text were small.

Review Comments to the Author

Reviewer #1: This study assessed the 20-year impact of tobacco price increases and increased tobacco control expenditure during 1998-2017 in Minnesota using a microsimulation model developed by the authors. Different from most of similar studies in the literature, this study include a broader array of outcome: smoking prevalence, disease events, mortality, smoking-attributable medical costs, productivity gains, and state cigarette tax revenues. This a well-written article with details of their models and simulation methodologies presented in the appendices. Although this study made many assumptions about their model parameters, it provided sensitivity analyses to explore the influence of a wide range of changes in model parameters. This study demonstrated that increasing tobacco prices and investments in tobacco control simultaneously is effective to reduce the harms of tobacco use and is cost-saving in the mid- to long-term. I only have several minor comments.

1. Page 7, the methodology of estimating the consequences of cigarette smoking was not explained sufficiently. More description is needed regarding whether and how an econometric model was developed to estimate the smoking-attributable medical costs for current smokers. According to Table S3.1 under the baseline scenario, smoking-attributable medical costs was $1.324 billion in 2017. It would be great if the authors can add some discussion to compare their annual SA medical cost estimates with similar studies in the literature.

Rather than adding additional technical detail to the text, we now refer readers to the places in these supplements where details can be found. We are aware of no other published estimates of smoking-attributable medical expenditures for the state of Minnesota. Due to the importance of medical cost inputs on the estimates of medical costs averted, we had already included results using alternate medical cost estimates that align better (on a per-person basis) with the national estimates provided in the 2014 Surgeon General’s Report. We now draw out these results in the Results of Sensitivity Analyses by adding the following sentence: “Using alternative cost estimates that align with those in 2014 Surgeon General’s report increased savings by about 35%.”

2. Page 9, lines 151-153: It is not clear whether the price increases mentioned here ($0.76, $0.67, and $3.10) reflect constant dollar or nominal term. For example, Line 152 indicates that in subsequent years after 1999, the federal excise tax on cigarettes increased by $0.67 per pack. However, during that period, the federal cigarette taxes increased 3 times: by $0.10 per pack in 2000, by $0.05 per pack in 2004, and by $0.61 per pack in 2009. The sum of these increases is $0.76 which does not match to $0.67.

Thank you for the careful read. We now clarify that these are not adjusted for inflation. There was a typo that listed the federal excise tax increases as $0.67 rather than $0.77. Please note that we calculate the total as $0.77 rather than $0.76 because our source shows the federal increase in 2009 to be $0.62 rather than the $0.61 stated in the reviewer’s comment.

3. Page 9, lines 155-157: “The difference in health … between this ITC + Price Scenario and the ITC scenario provides estimates of the impact of tax increases alone”. This sentence is not correct because this difference also reflects the impact of the price increase due to the Minnesota tobacco lawsuit settlement.

Thank you. We have revised the sentence. It was meant to convey that the estimates exclude ITC (now ITCI) effects, but that was not clear.

4. Page 11, lines 188-189. The reduced numbers of smokers in 2017 (7,434 for youth and 138,123 for adults) due to ITC do not add up to the number shown in Table S3.4 (146,402).

Thank you. The numbers in the tables were correct. The number in the text for adults was misread – it was another similar number in the underlying spreadsheet that was used when completing the text.

5. Page 11, lines 201-205; page 12, line 206 and lines 211-203: All the numbers presented in these sentences do no match to the numbers shown in Table 1 and Tables S3.4.

As above, numbers in tables were correct. We apologies for the careless entry of numbers in the text.

6. Page 12, line 207 and line 215: Add “(data not shown”) after “$784 million”, and similarly after “$3.20 billion”.

Another row has been added to provide these numbers in Table 1.

7. Page 13, the last row “Base case”: the number shown in the last column does not match to the corresponding cell in Table 1.

Thank you. This appears to be rounding error causing the numbers to differ by 1. This level of precision is no longer visible after following guidance from the editor to round all results to a justifiable number of significant digits.

Reviewer #2: This manuscript presents the findings of a microsimulation to assess the impact of tobacco control policies, including tax increases and increased expenditures on tobacco control, in Minnesota during the twenty year period spanning 1998-2017. The study found that that increased expenditures on tobacco control prevented 37,421 cancer, cardiovascular, diabetes and respiratory disease events and 3,839 deaths over 20 years. Additionally, increased prices prevented an additional 14,702 additional events and 1,662 deaths. The authors conclude that both the net increase in tax revenues and the reduction in medical costs were greater than the additional investments in tobacco control, and that states can pay for initial investments in tobacco control through tax increases and recoup those investments through reduced healthcare expenditures.

This manuscript is generally well written, the analytic approach is free from any major methodological pitfalls, and would make a meaningful contribution to the scientific literature. Nonetheless, the manuscript could be improved, including more prominent recognition of various limitations of the analysis. Specific recommendations for revision are as follows:

1. Abstract. Page 2. Introduction. The first sentence of the abstract understates the considerable body of scientific evidence on the efficacy of tobacco control programs; specifically, the language needs to more clearly state the direction of the effect. They don’t just “influence” tobacco use and related health and economic outcomes – they reduce these outcomes.

Thank you. The sentence now reads “Tobacco control programs and policies reduce tobacco use and prevent health and economic harms.”

2. Introduction. Page 4. Third paragraph. The third sentence references “prior initiatives” and “significant policy change during this period.” However, it’s not clear to a lay reader what this means, particularly those without tobacco control expertise. A sentence should be added that clearly articulates what policies were actually implemented.

This suggestion helps to make the text more concrete. We have divided the sentence into two so we can provide sufficient detail. It now reads: “Part of this trend arose from earlier tobacco control policies including the nation’s first state-wide clean indoor air law in 1975, the nation’s first state-funded tobacco control program in 1985, and four state tax increases from 1985 to 1992. Significant policy changes after 1997 added to the trend, including strengthening the 1975 clean indoor air law to include all indoor public places, strengthening youth access laws, further increases in cigarette taxes and substantial increases in tobacco control expenditures.”

3. Introduction. Page 4. Since the manuscript only focuses on tobacco control program expenditures and price increases (i.e. and not other specific proven strategies such as smoke-free policies), it would be helpful to better set the stage for inclusion of these variables in the Introduction, including the focus on increased price. One way to accomplish this would be to reinforce that tobacco control expenditures support a variety of efforts, including state and community programs that influence policy adoption, and that increasing price has previously been noted as the single most effective intervention for reducing consumption. That way you better set the stage for the reader around the measures focused upon in the manuscript.

Thank you for the suggestion. We believe this helps to focus the introduction as it leads into the methods. The penultimate paragraph of the introduction now ends with: “Large tobacco price increases are the most effective known strategy to reduce tobacco use. Tobacco control expenditures support the state’s free quit line, anti-tobacco media campaigns, tobacco education, communication, and community collaborations. Increased expenditures also support initiatives to reduce the harms of commercial tobacco to American Indian communities, and education and communication activities likely contributed to passing local Tobacco 21 laws and local restrictions on the sale of menthol cigarettes.”

4. Introduction. Page 5. First paragraph. The last two sentences of the Introduction section are implication and discussion points, not Introductory content. They should be deleted here, and reinforced in the Discussion section.

We agree. The last sentence was already well-covered in the Conclusions and therefore we deleted it entirely. The second to last section has been moved to the Discussion section.

5. Methods. Page 5. Demographics and smoking status. It’s not clear what specific age groups from the “Minnesota population in 1997” the data were adjusted to – was it all individuals, those 18+, or something else?

We have revised the sentence to clarify that this includes all ages.

6. Methods. Page 6. The paragraph beginning “we estimated adult cigarette smoking” lacks necessary precision for the reader to understand the rationale for the employed analytic approach. For example, it’s not clear why 1996 and 1997 data were combined (sample size issue?). Additionally, it’s not clear what is meant by “calibrating the initial smoking probabilities”. Furthermore, it’s not apparent how the 18-24 year old age group, for which there was insufficient sample, could be predicted using estimates for 25-year olds, which reflects only a single age year; did you mean some range that began with 25 as the lower bound? If sample size were an issue, it would seem that you could consider adding multiple years of data to afford sufficient sample for 18-24 year olds, which is a critical smoking initiation demographic. If you can’t this needs to be duly noted as a limitation of the analysis.

We have clarified that the two years were combined to add precision, particularly in estimating cessation rates. We would need to combine many years to obtain an adequate sample of surveyed Minnesotan’s to obtain statistically more precise estimates by sex for ages 18-24. However, with changing smoking behaviors during this time, using multiple years would not necessarily provide a more accurate estimate. We have added to the methods section the following sentence: “This assumption could understate peak adult prevalence during the lifetime but provides a reliable estimate of prevalence prior to the ages of high harms from smoking-attributable chronic disease.” We have added to the limitations part of the Discussion section the following sentence: “Our assumption on tobacco use for adults less than 25 years of age would cause us to simulate too few former smokers if peak adult prevalence occurs before age 25.”

7. Methods. Page 7. Consequences of Cigarette Smoking. More clarity needs to be provided in the first paragraph of this section to articulate what smoking-attributable diseases were included. The current framing suggests it could be some, but not all. To clarify, it might be more prudent to place the onus of inclusion on outcomes found to be significantly associated with smoking in the 2014 Surgeon General’s Report, rather than focusing on SAMMEC, which is no longer in existence.

We modified the sentence to note that cancers, cardiometabolic and respiratory diseases are included and we now refer readers to places in supplement S1 where each disease is listed. Please note that while SAMMEC no longer exists as an electronic warehouse on CDC’s webpage, it does exist in the 2014 Surgeon General’s Report as Chapter 12 Smoking-Attributable Morbidity, Mortality, and Economic Costs . The CDC’s Office of Smoking and Health refer to the 2014 SGR as the updated SAMMEC.

8. Methods. Page 8. It’s unfortunate that the authors could not include smoke-free policies in the analysis, which was undoubtedly a contributor to both declining secondhand smoke exposure, as well as reduced smoking, in Minnesota during the assessed period. This needs to be stated as limitation of the manuscript (along with other proven strategies that couldn’t be included). Additionally, the text in this paragraph of the Methods should use the more commonly recognized term of “secondhand smoke exposure” instead of “secondary smoke exposure”. It’s also not clear what is meant by the statement “the analysis plan was not pre-registered.”

We have changed the text language to secondhand smoke and the limitations portion of the Discussion section now notes the exclusion of the benefits of reducing secondhand smoke from the estimates.

9. Methods. Page 8. It’s strongly recommended that the authors identify another abbreviation than “ITC” for “Increased Investments in Tobacco Control.” The abbreviation “ITC” has long been associated with the International Tobacco Control Survey in the tobacco control field, and thus, using this abbreviation may cause confusion when findings are reported elsewhere or taken out of the direct context of this study. Instead, authors could consider something like Increased Tobacco Control Investments, or “ITCI”.

Thank you for the suggestion. We have adopted ITCI throughout to prevent confusion and prevent readers from being distracted by a familiar acronym being used for a different concept.

10. Methods. Page 9. It’s questionable whether it is safe to assume that one policy does not impact the relative effectiveness on smoking behaviors of the other policy. Increased expenditures for tobacco control programs include state and community interventions, as outlined in CDC’s Best Practices for Comprehensive Tobacco Control Programs. Price increases are considered a product of those programs. As such, increased expenditures for programs would be associated with increased momentum around pricing strategies like excise taxes. Since there’s not much else the authors can do post hoc, at the least this should be duly noted as a limitation of the analysis.

We had this concern in mind when reviewing literature to obtain policy effectiveness estimates (the elasticities). The literature estimates which provide the effect sizes for each policy in the model are estimated while controlling for the effects of other policies on tobacco behavior. Thus, the simulation results are independent of each other to extent that the statistical control for contemporaneous policies in estimating policy effects in the literature was successful.

11. Discussion. Page 14. It is strongly recommended that the authors not pit individual interventions against one another. The framing around investments being more impactful than price would surely be used by opponents of cigarette excise taxes to state that they aren’t needed because they aren’t as effective. The major take home here is really that both are effective and important as part of a comprehensive approach to tobacco prevention and control at the state level.

We appreciate the concern and have clarified the text on page 14 to make it clear that the results pertain to the size of price increase and size of expenditure increases implemented in Minnesota. Clearly, the impact of one or the other will depend on the magnitude of each increase. We can’t prevent a disingenuous tax opponent from misusing the information that is provided in the results section. Funding tobacco control, has proven more politically difficult than raising tobacco taxes (even when funding was endowed by tobacco settlements, states raided the endowments). The results show that funding tobacco control is an important component. The Conclusions make the case for the price increases and additional investments in tobacco control being complimentary in a comprehensive tobacco control strategy.

12. Discussion. Page 15. The authors appropriately note that tobacco industry marketing efforts could counter the impact of tobacco control investments. However, this point could be made clearer for lay readers if the most recent data from the Federal Trade Commission were cited. In 2017, $9 billion was spent on advertising and promotion of cigarettes— more than $1 million every hour.

Thank you for the suggestion. We have added the best-available estimates of tobacco industry marketing expenditures in Minnesota to the text: “The tobacco industry spent an average of $165 million on tobacco marketing in Minnesota each year from 1998 to 2017 as estimated from national expenditures and state tobacco sales.”

13. Discussion. Page 16. The manuscript needs a paragraph that clearly articulates the limitations of the employed analytic approach, including several of the items noted previously in this review (e.g. lack of smoking estimates for 18-24 year olds, omission of full scope of policies such as smoke-free, assumption about independent effect of price and tobacco control program expenditures, etc).

Substantial new text has been added that covers most of these topics. However, we do not agree that omitting tobacco policies that were outside our scope is a limitation of your results.

14. Discussion. Another key point to reinforce in the Discussion is that this analysis only accounts for cigarette smoking, not all tobacco products, including other combustible products, smokeless tobacco products, and e-cigarettes. During the later part of the assessed period, particularly from 2011-2017, massive increases in the use of e-cigarettes were observed among youth – both nationally and in Minnesota. Therefore, this model doesn’t account for the full scope of tobacco products being used, as well as associated health risks/costs. Cigarette smoking is certainly responsible for the overwhelming burden of death and disease caused by tobacco use, but these other products are not without risk. Accordingly, this needs to be explicitly stated as a limitation of the analysis, and ideally, a reference to the diversification of the tobacco product landscape should also be made in the narrative elsewhere in the manuscript.

Thank you for the suggestion. This has been added to the expanded limitations portion of the discussion section. Also, at the beginning of the methods section, we now note that the model does not cover the complex arena of multiple tobacco product use.

15. Discussion. Another essential point that is briefly referenced, but should be more prominently focused upon, in the manuscript is that a comprehensive approach is warranted. There is no single panacea, and for several decades, tobacco control has most effectively functioned through a comprehensive lens that addresses the various drivers influencing initiation and cessation. As currently framed, the manuscript could be misinterpreted as suggesting that either expenditures and price are effective alone. It’s also important to reinforce that it’s not just making the expenditures, but what those dollars are actually used for; large amounts of dollars allocated to ineffective initiatives or strategies will do little for preventing initiation and reducing consumption at the population level. The expenditures needs to be focused of proven strategies for which evidence of efficacy and cost-effectiveness have been established.

We have added text that creates a better picture of a comprehensive approach in response to item #3 above. In addition, the conclusion now states that “Comprehensive state tobacco control programs that combine price increases, investments in tobacco control and policies that prevention initiation, and help smokers who want to quit have been found to be the most effective strategy for reducing the harms of tobacco.”

Reviewer #3: This is an important paper with findings relevant to continuing policy decisions about the investment of millions of tobacco settlement dollars. The economic analyses are detailed and comprehensive, and provide the basis for important policy discussions and conclusions. As such, the strength and details of these analyses are needed. However, this leads to a key question: what is the primary audience of this paper? The decision appears to be health policy makers, and not health economists. The details in the Supplemental Files are dense but important, and if more were included in the paper it would become more of a health economist paper. However, as written, many of the critical technical details of the model, parameter definitions and selection, sensitivity analysis, and technical aspects of the results are left to the Supplemental Files. Even so, the language of the paper and topics discussed still focus too much on economic issues rather than presenting key findings most relevant to the policy makers. For example, while noted in the discussion that "estimating return on investment for increased ITC was not a study goal..." (lines 290-291), the findings of a positive return on investment even during the limited time horizon is very important, and should not be buried deep in the Discussion. Similarly, the sensitivity analysis is important to these potential return on investment discussions and conclusions, but as presented, these analyses are written more for economists, and as such, will be too confusing to most policy makers. The critical link between the sensitivity analyses and return on investment conclusions needs to be clearly defined when introducing them (line 226). Also, while the Figures (1 and 2) are good at communicating the impact of ITC and Price, the specific data in Table 1 and Supplemental Files S3 are not presented well in the text for policy makers, and especially not summarized well in Abstract. The focus is on deaths and SA incidents averted; however, the bottom line for many policy makers is dollars saved. The medical cost savings and productivity gain estimates are unique to these analyses and should be given primary emphasis, along with the resulting return on investment calculations based upon them.

We have struggled with balancing implications for policy makers of the secondary economic findings with the original intent of the analysis. We believe that scientific integrity requires that we present the results in a manner that is consistent with the research plan and the manner in which the results were revealed to us. The intent of the analysis was to delineate what the contributions of cigarette price increases and tobacco control expenditures to Minnesotan’s health and economic well-being have been. We did not set out to determine if investments in tobacco control were cost-effective or produced a positive return on investment in a 20-year timeframe. While we agree that the secondary results that compare the investments in ITCI to their economic impact are likely to be the most compelling to policy makers, we feel it would be incorrect to focus the article on secondary results that happened to be very compelling when we would not have done so had we found that the ITCI did not yield a positive ROI within 20 years. We feel that this journal, with its focus on transparency and methodological rigor, is the proper venue for a paper presented in this manner. 

Along the same lines, while we appreciate that a paper focused on ROI could make a crisp, succinct policy paper, our experience is that decision makers, especially those who champion a policy, want to be able to say that the policy they advocate has a sound basis, and that may require a detailed read and opinion from a trusted technical advisor. At the same time, few health economists would be interested in the technical merits of a study that had no practical implications. For these reasons, the delineation between a paper written for decision makers and one written for economists is not clear.

Therefore, we agree with editor’s advice to provide the additional detail on methods being sought by all reviewers while reinforcing the policy implications in the paper. Where possible, we now refer readers who may be looking for methods detail to specific places in the supplementary materials and we have added statements in the introduction, results, and discussion to better highlight the policy implications.

(Reviewer:) Also, the wealth of literature review, technical discussions, and econometrics in Supplemental Files S1 and S2 are very impressive. As noted above, if the primary audience of the paper is health policy makers, then these details need to stay there. However, the descriptions of the technical approaches in the main paper should be reviewed and considered: are they communicating the technical quality of the work presented in the Supplemental Files to the policy makers as effectively as possible? In many places, the reader should be specifically referred to the excellent presentation of the issues and parameter selection criteria in S1 and S2.

Thank you for recognizing the substantial background work presented in the supplements that made the analyses possible. We now refer readers to the supplements in multiple places and better draw readers’ attention to the type of detail that can be found in the supplements. For example, the discussion of the ICTI + price scenario, we now state: “These elasticities were drawn from literature with careful consideration of which studies provide the most appropriate price elasticity estimates as inputs to the simulation model, and on how the results of multiple studies should be combined (see supplement S2).”

More detailed comments:

Lines 43-48: I believe this is the only microsimulation model -- and the others, particularly the Levy model is a relatively crude macrosimulation model. This distinction should be emphasized.

We now note in the discussion section that differences in the structures of the models may contribute to differences in their results. Tobacco microsimulations are becoming more common, particularly in the literature on the impact of e-cigarettes on the population harms of tobacco. To our knowledge none have the detail in outcomes of the model used for this analysis, and therefore we focus on that distinction in this paper. The Levy model is well-validated and has provided import insights into tobacco policy for many years.

Lines 59-65: the potential of these findings to provide data needed for cost-effectiveness and return on investment calculations should be introduced and emphasized.

We have added to the introduction the sentence: “The results demonstrate the potential for health and economic gains in states that have not acted aggressively against tobacco.” We hesitate state that the results from the size and timing of price increases and tobacco control expenditure increases in Minnesota for a specific time period can be directly used in cost-effectiveness analyses.

Line 70: the details in literature review supporting selection of data inputs, policy parameters, and details in model should get more emphasis here.

This sentence now reads: “The simulation model, data inputs, and policy parameters are described in supplements S1 and S2, including details on the use of databases and literature to inform the model.” We now call attention to the supplements in additional places throughout the text.

Lines 128-130: The discussion of clean indoor air legislation should be revised. The specific impact on CIA on health effects of reduced exposures to secondhand smoke are not included (as noted); however, the policy impact of CIA on other outcomes by default are included in the ITC. While these are not specifically included in the model parameters, their impact is captured in the overall results. This should be noted.

We agree that the impact of clean air laws can be attributed to ITC to the extent that communications funded by ITC contributed to passing clean-air laws. However the studies that inform the impact of ITC on tobacco used behaviors in the model control for the existence of clean air laws and therefore provide an estimate of the effects of ITC on tobacco use that is independent of the impact clean air laws.

Lines 131-137: The details of annual tobacco control investments are provided in Supplemental Files. Guide readers to these details. This also could apply to other data included in the model and only shown in detail in Supplemental Files; i.e., annual excise tax levels and average annual retail cigarette prices.

Thank you. This is done.

Lines 167+: as noted above, the important relevancy of these simulations for explaining return on investment after adjusting for possible biases and reduced parameter estimates for impact of ITC or price, i.e., expenditure and price elasticities, etc.

These policy implications of the simulations reported in this section are prominent in the Conclusions.

Line 192: "...due to lagged effect of prior investments." will not be understood my almost all readers as being primarily due to two model decisions: first, only one time impact of price increases applied at time of tax increase (standard econometrics) for adults, but impact on youth extended; and second, the use of cumulative funding with 25% discount (a la Farrelly et al models). Maybe this is too technical a point to discuss in results, but might merit being raised in discussion.

We agree and have deleted this phrase. In an early draft we had additional explanation for this phrase and decided it was too detailed for most readers and was distracting.

Line 209: the detailed impacts are shown in S3, but more of them could be shown in an extended Table 1. Additionally, details on annual ITC should be linked with results in S3 to highlight return on investments. Also, S3 could show more details on annual state tax revenues under Baseline Scenario as well as under actual tax increases. Finally, it would be outstanding to see a combined return on investment table combining tax revenues, ITC investments, and SA medical costs. Such a table could have very important policy implications.

We have added, and refer readers to, two new tables in S3 the show changes in tax revenues, ITC expenditures, medical costs and productivity effects side-by-side. We feel that adding a time dimension to Table 1 to show the detail provided in S3 would be very challenging and would distract from the bottom line results.

Table 1: something does not look right. I did not search out all the details in Supplemental Files, but the deaths do not look right. SA cancers (incidents) and SA deaths are almost identical; however, SAMMEC estimates are that only about one-third of SA deaths are due to cancer. This needs to be checked, since SA deaths should be much higher than SA cancer incidents and deaths.

These estimates are correct. Only a fraction of cancer cases are fatal. We have modified the label in Table 1 to make it clear that these are cancer cases. 

Figure 2 (lines 217-224): emphasis on percent changes, but policy impact in $s only shown in Table 1. Figure could be 2a (current) and 2b with numbers (number of youth prevented, increase number of adult quitters, SA incident cases prevented, SA deaths averted, and most importantly, SA costs saved (in $s), productivity gains (in $s) and (to be computed) total savings (SA medical averted and productivity gains).

Figure 2 provides a different view on the results than provided in Table 1, with an emphasis on relative change that provides a glimpse of what may occur in future years. It is not clear to us what would be gained by repeating the numbers of Table 1 in a figure. We chose to display them in a figure so readers could easily read the numerical estimates and because the different outcomes are on much different scales and thus do not lend themselves being displayed a single chart.

Lines 225-237: as noted above, the impact of sensitivity analyses on return on investment calculations needs more emphasis. Lines 233-236 briefly discuss these findings; but, these calculations need much more emphasis and should be provided in greater detail; maybe even added as additional columns in Table 2.

Please see discussion above.

Lines 239-255: this discussion is relevant, but may not strike the right balance. Make it clear from the first sentence that both ITC and price increases are needed: they are complimentary in timing and level of impacts produced.

The first paragraph of the Discussion has now been split into two paragraphs, with the impacts on tax revenues added to the first paragraph. The Conclusion makes the case that the two policies are complimentary.

Lines 256-267: discussion of Levy et al model results relevant, but this is a microsimulation model that differs in many ways from their model beyond parameter estimates.

We now note in the discussion section that differences in the structures of the models may contribute to differences in their results.

Lines 287-296: discussion of uncertainties and imprecisions in model parameters and inputs is always relevant. The details in S1 and S2 provide very good discussion of how parameters and model specifications were done. Point readers to these discussions and reviews. Yes, uncertainties about elasticities, especially about ITC, limit model predictions. Thus, the need for good sensitivity analyses (as was done). Nevertheless, the positive return on investment calculations were found, as well as the positive revenue gains for tax increases (which need more emphasis with more specific numbers provided, even thru this has been documented in many prior econometric analysis).

Additional referrals to S1 and S2 have been added in the text as noted above. The tax revenue results have been added to the first paragraph of the discussion. Tables comparing revenues and the other economic measures have been added to S3.

FINALLY, it is suggested that future analyses estimate the even greater impacts (on all outcome measures, but particularly SA cost savings) if ITC levels would have been higher (like full BP levels or even 50% of BP level) or would not have been reduced in 2004.

Thank you for the suggestions. We agree; estimates of full potential of these policies are needed.

---

## [Editor Report · Decision Letter 1]

24 Feb 2020

PONE-D-19-28016R1

The 20-year impact of tobacco price and tobacco control expenditure increases in Minnesota, 1998-2017

PLOS ONE

Dear Dr. Maciosek,

Thank you for submitting your manuscript to PLOS ONE. After careful consideration, we feel that it has merit but does not fully meet PLOS ONE’s publication criteria as it currently stands. You have done a nice job of responding to the reviewers.

There is only one small change left.  In the Introduction, you say that "Large tobacco price increases are the most effective known strategy to reduce tobacco use.”  While this is a commonly made statement, as you note elsewhere in the paper, how effective a tax increase is depends on the size of the tax increase.  For example, a smokefree law could have a bigger effect than a small tax increase.  Please replace "the most effective" with "an effective" and make any other needed wording adjustments to the paper to avoid overstating the importance of tax increases.

We would appreciate receiving your revised manuscript by Apr 09 2020 11:59PM. To enhance the reproducibility of your results, we recommend that if applicable you deposit your laboratory protocols in protocols.io, where a protocol can be assigned its own identifier (DOI) such that it can be cited independently in the future. For instructions see: http://journals.plos.org/plosone/s/submission-guidelines#loc-laboratory-protocols

We look forward to receiving your revised manuscript.

Kind regards,

Stanton A. Glantz

Academic Editor

PLOS ONE

---

## [Author Response · Author response to Decision Letter 1]

26 Feb 2020

Editor Comments:

There is only one small change left. In the Introduction, you say that "Large tobacco price increases are the most effective known strategy to reduce tobacco use.” While this is a commonly made statement, as you note elsewhere in the paper, how effective a tax increase is depends on the size of the tax increase. For example, a smokefree law could have a bigger effect than a small tax increase. Please replace "the most effective" with "an effective" and make any other needed wording adjustments to the paper to avoid overstating the importance of tax increases.

Response: 

We agree that this is a contradiction. We have made the suggested change in the introduction and removed characterizations of the effect size of tobacco taxes (e.g. ‘large’, ‘substantial’) from the results section.

---

## [Editor Report · Decision Letter 2]

28 Feb 2020

The 20-year impact of tobacco price and tobacco control expenditure increases in Minnesota, 1998-2017

PONE-D-19-28016R2

Dear Dr. Maciosek,

We are pleased to inform you that your manuscript has in been judged scientifically suitable for publication and will be formally accepted for publication once it complies with all outstanding technical requirements.

With kind regards,

Stanton A. Glantz

Academic Editor

PLOS ONE
---

## [Editor Report · Acceptance letter]

5 Mar 2020

PONE-D-19-28016R2 

The 20-year impact of tobacco price and tobacco control expenditure increases in Minnesota, 1998-2017 

Dear Dr. Maciosek:

I am pleased to inform you that your manuscript has been deemed suitable for publication in PLOS ONE. Congratulations! Your manuscript is now with our production department. 

With kind regards,

on behalf of

Professor Stanton A. Glantz 

Academic Editor

PLOS ONE